# Research on stroke patients' perception of recurrence risk: A scoping review protocol

Shanshan Zhu[1], Xueting Sun[1], Xin Guo[1], Meiqi Xu[2], Dingding Li[1], Shuaiyou Wang[1], Yage Shi[1], Chenjun Liu[1], Hongru Wang[2], Huimin Zhang[1] *

1 School of Nursing, Xinxiang Medical University, Xinxiang, Henan, China, 2 Nursing Department, Fifth Clinical College of Xinxiang Medical University, Xinxiang Henan, China

* 35290915@qq.com

**Data Availability Statement:** No datasets were generated or analysed during the current study. All relevant data from this study will be made available upon study completion.

## Abstract

### Introduction

Stroke, a major global cause of death and disability, has a high recurrence rate that significantly affects patients' physical, psychological, and economic well-being. Despite the importance of health risk perception in preventive measures, most stroke patients struggle to accurately assess the risk of recurrence. Current research on stroke recurrence risk perception is still exploratory, with a lack of systematic understanding of the influencing factors. This study aims to comprehensively analyze the current state of stroke recurrence research and the factors that influenced recurrence and assess the effectiveness and limitations of various assessment tools to guide future research and intervention strategies.

### Methods and analysis

This scoping review will follow Arksey and O'Malley's methodological framework as well as the updated scoping review methodology guidance by the Joanna Briggs Institute (JBI). Review results will follow the Preferred Reporting Items for Systematic Reviews and Meta-Analyses extension for Scoping Reviews (PRISMA-ScR) guidelines. The search strategy will be developed via keywords, such as stroke, recurrence risk perception, and influencing factors. We will systematically search seven English databases, PubMed, CINAHL, Web of Science, Embase, Cochrane Library, PsycInfo, and MEDLINE, as well as four Chinese databases, CNKI, Wanfang, VIP, and the China National Knowledge Infrastructure for Biomedical Literature. Studies published in both English and Chinese will be included. Data will be extracted via a standardized form and summarized through quantitative (frequency) and qualitative analyses (narrative synthesis). Furthermore, the findings will be reported.

### Ethics and dissemination

Since this review involves collecting data from existing literature and does not involve human participants, ethical approval is not required. Research findings will be disseminated through conference presentations and publications in peer-reviewed journals.

**Funding:** Funding: This work is supported by the [Key Scientific Research Project of Higher Education Institutions of Henan Province] (grant number: 23B320002), [Henan Province Higher Education Teaching Reform Research and Practice Project] (grant number: 2023SJGLX236Y), and [Henan Province Graduate Education Reform and Quality Improvement Project] (grant number: YJS2024JC27).

**Competing interests:** The authors have declared that no competing interests exist.

## Registration details

This protocol has been registered on the Open Science Framework (OSF). Relevant materials and potential following updates are available at https://osf.io/7kq5t.

## Introduction

Stroke is the second leading cause of death globally and third leading cause of combined death and disability as measured in disability-adjusted life years (DALYs) [1–4]. It severely endangers physical and mental health, and is the leading cause of death and disability among adults in China. It is characterized by high incidence, recurrence, disability rate, and mortality and significant economic burden [5]. Particularly, its high recurrence rate is concerning. A meta-analysis examined the cumulative risk of stroke recurrence over the past decade (2009–2019) and revealed that the recurrence rate was 7.7% after three months, 10.4% after one year, and 12.9% within 10 years [6]. Ng et al. [7] found that disease recurrence adversely affected patients' physical and psychological health, substantially increased the risk of unemployment, and decreased the likelihood of returning to work. Furthermore, recurrence may lead to cognitive impairments and an increased risk of post-stroke dementia [8]. These factors significantly exacerbate medical and economic burdens for the patients and their families [9]. Therefore, assessing the risk of recurrent stroke and effectively managing the risk factors is a critical component of secondary prevention [10].

Health-related theories, such as the Health Belief Model [11] and Protection Motivation Theory [12], propose that an individual's perceived health risk serves as a foundation for undertaking preventive behaviors. The Risk Compensation Theory suggests that individuals are more likely to engage in risky behaviors when they are unaware of the risks involved [13]. This indicates that perceived risk/susceptibility serves as a fundamental driver for patients to proactively engage in health behaviors and disease management [14]. Aycock et al. suggest that overestimating the risk of contracting a disease can lead to chronic stress among individuals, which may potentially cause additional physical harm. Conversely, underestimating such risks may result in lower engagement and compliance with treatment regimens [15]. Therefore, objectively and accurately assessing the perceived risk of recurrent stroke in patients is crucial. Perceived recurrence risk refers to awareness of early warning signs related to disease recurrence and severity and behavior-related and disease-related risk factors [16].

Recent research has examined the concordance between self-perceived and objective recurrence risks among patients after a stroke. However, findings indicate that most stroke patients are unable to accurately perceive their risk of recurrence. Furthermore, research on the factors that influence recurrence risk perception is still exploratory, and effects of the existing factors have not reached a consensus [17–19]. Studies on risk perception in the field of disease should be approached from a complex and dynamic perspective, as individual responses to disease risks are influenced by a multitude of factors [20]. In most cases, perceptions of risk are derived from knowledge of life experiences and disease characteristics and are also profoundly affected by an individual's psychological states [17, 21], the risk information they receive [22], and their broader social environment [23].

Recently, numerous tools have been developed and designed to help clinicians accurately evaluate and assess stroke patients' perceptions of recurrence risk. These assessment tools vary in their development background, characteristics, advantages, disadvantages, and applicable scopes [24]. They also differ regarding the timing of the assessment, evaluation indicators, and

methods. Some tools specifically focus on the likelihood of and susceptibility to recurrence and often use single- or multiple-question assessments to measure the risk of recurrence over the next 1, 10, or 20 years. Responses typically utilize yes or no answers, Likert scales, visual analog scales, or numerical rating scales [25]. Therefore, to comprehensively and accurately assess the level of stroke patients' perception of recurrence risk, this study aimed to conduct a scoping review of the tools used to assess the risk perception of recurrence in stroke patients and identify the application effectiveness and limitations of the different tools in various contexts.

Despite increasing studies on stroke recurrence risk perception, a systematic understanding of its current status and influencing factors is lacking. A scoping review is an efficient method to rapidly examine progress in a specific research area, summarize the existing studies, and identify their limitations. Accordingly, this study employed scoping review guidelines as a methodological framework to synthesize and analyze the current research status and influencing factors. We aim to provide clear directions for future studies and foster the development and implementation of effective interventions for stroke patients' recurrence risk perception.

## Objectives and research questions

This review aims to investigate the current state of research on stroke patients' perceptions of recurrence risk and identify relevant factors. We aim to summarize existing literature in this domain. Specifically, this review will address the following research questions:

1. How many studies have been conducted on the perception of recurrence risk among stroke patients till date, and what types are they?

2. What are the factors that influence stroke patients' perception of recurrence risk?

3. What are the assessment tools for stroke patients' perception of recurrence risk?

4. How do stroke patients describe their experience of developing a perception of recurrence risk?

5. How do stroke patients explain the impact of their perception of recurrence risk on their health behaviors and lifestyle?

## Methods and analysis

### Study design

This scoping review will follow Arksey and O'Malley's proposed methodological framework [26] and incorporate the updated scoping review methodology guidance provided by the Joanna Briggs Institute (JBI) [27]. The Preferred Reporting Items for Systematic Reviews and Meta-Analysis extension for Scoping Reviews (PRISMA-ScR) checklist [28] will be followed to improve scientific rigor, as seen in the supporting information (S1 Checklist). This protocol has been registered through the Open Science Framework (https://osf.io/7kq5t).

### Eligibility criteria

**Inclusion.**   Inclusion criteria were based on the study population, concept, context, and type of evidence source.

1. *Population*. Stroke patients aged 18 years and older, including those with ischemic and hemorrhagic strokes, without restrictions regarding gender, ethnicity, geographical location, or time since diagnosis.

2. *Concept*. Studies that explored stroke patients' perception of recurrence risk, including risk awareness (understanding and knowledge of the likelihood of recurrence), attitudes (emotional responses towards the possibility of recurrence), and management (actions and behaviors to manage and mitigate recurrence risk) and the influencing factors (demographic, psychological, social, and clinical factors).

3. *Context*. Studies conducted in community health centers, hospitals, and home settings.

4. *Type of evidence source*. Studies that were available in full-text publications and as research papers, irrespective of research methodology or design, such as quantitative studies (interventional, cross-sectional, and longitudinal), qualitative studies, or mixed-methods research. All publications in Chinese and English will be included.

**Exclusion.**

1. Documents that did not adequately address the research questions (e.g., guidelines, reviews, research proposals, and government documents).

2. Incomplete articles or those with inaccessible full text.

## Information sources and search strategy

**Information sources.**   Searches will be conducted across seven English-language databases (PubMed, CINAHL, Web of Science, Embase, Cochrane Library, PsycInfo, and MEDLINE) and four Chinese-language databases (CNKI, Wanfang, VIP, and China Biomedical Database). The search period will be from the inception of each database to December 2024. In addition, we will search the reference lists of the relevant studies and gray literature to achieve a comprehensive retrieval. Furthermore, before the final analysis, we will re-run the analysis to identify any new relevant studies.

**Search strategy.**   The search strategy was developed via MeSH terms and keyword combinations. In addition, a manual search will be conducted via a reference tracing method to identify potentially missed publications. The comprehensive literature search strategy was determined through discussions between an experienced information specialist (Zhang) and the first author (Zhu). Search terms that include all the other relevant keywords, subject headings, and free-text terms, such as "stroke*," "perceived risk of recurrence," "influencing factors," and "current situation," will be used to identify the relevant studies. According to the PRISMA-ScR checklist and explanation [28], a complete search strategy should be provided for at least one electronic database, as demonstrated by the PubMed search strategy (presented in Table 1). Owing to the iterative nature of the scoping review [26], the initial search results will be assessed, and any required improvements will be considered. Any changes to the protocol will be reported.

**Study screening.**   All retrieved literature will be managed via EndNote X9 software and duplicates will be removed. Two researchers (SZ and XS) will independently screen the titles, abstracts, and full texts, exclude the irrelevant studies, and ensure included studies meet the inclusion criteria. Following the preliminary screening, a further detailed review of the full texts will be conducted based on the inclusion and exclusion criteria. In case of disagreements, a third researcher (HZ) will be consulted to reach a consensus. Additionally, the reference lists of the included literature and relevant reviews will be checked to identify potentially relevant articles, which will also be screened via the same process. The entire literature retrieval and screening process will be detailed in the final scoping review and presented in a PRISMA flowchart [29]. Results of this stage will be presented in a flow chart (Fig 1).

Table 1. Search strategy example in PubMed as of 19 May 2024.

| # | Searches | Results |
|---|---|---|
| 1 | stroke[MeSH Terms] | 181595 |
| 2 | (((((((((stroke[Title/Abstract]) OR (strokes[Title/Abstract])) OR (stroke people[Title/Abstract])) OR (stroke peoples[Title/Abstract])) OR (stroke patient[Title/Abstract])) OR (stroke patients[Title/Abstract])) OR (poststroke[Title/Abstract]))) OR (stroke survivor[Title/Abstract])) OR (stroke survivors[Title/Abstract]) | 338007 |
| 3 | 1 OR 2 | 377830 |
| 4 | recurrence[MeSH Terms] Filters: from 1000/1/1–2024/4/30 | 203749 |
| 5 | (((((recurrence[Title/Abstract]) OR (recurrences[Title/Abstract])) OR (recrudescence[Title/Abstract])) OR (recrudescences[Title/Abstract])) OR (relapse[Title/Abstract])) OR (relapses[Title/Abstract]) | 567406 |
| 6 | 4 OR 5 | 687554 |
| 7 | risk perception[MeSH Terms] | 29052 |
| 8 | (((((((((((risk perception[Title/Abstract]) OR (perceived risk[Title/Abstract])) OR (Perceived susceptibility[Title/Abstract])) OR (Perceived vulnerability[Title/Abstract])) OR (risk awareness[Title/Abstract])) OR (Risk belief[Title/Abstract])) OR (Risk cognitions[Title/Abstract])) OR (attitude to risk[Title/Abstract])) OR (risk assessment[Title/Abstract])) OR (risk analysis[Title/Abstract])) OR (risk evaluation[Title/Abstract]) | 128320 |
| 9 | 7 OR 8 | 154913 |
| 10 | 3 AND 6 AND 9 | 97 |

**Data extraction.**    Based on the research objectives and questions, we convened a scoping review team meeting and developed and piloted a data extraction form in Excel (Microsoft Corp.). This form has been designed to comprehensively and adequately capture the relevant information. Furthermore, it may also be refined and updated during the review process. Data extraction will include the following items: (1) first author of the study (year of publication), (2) country/region, (3) study population/sample size, (4) research method/design, (5) research topic, (6) general condition of the patients, (7) data analysis methods, (8) assessment tools, (9) level of recurrence risk perception, and (10) influencing factors. To systematically identify the key factors that influence stroke patients' perception of recurrence risk, this study will employ an inductive thematic analysis method [30]. First, we will extract and summarize the three main levels related to risk perception: the individual (psychological state, emotional reactions), family and social (family support, socioeconomic status), and clinical and medical levels (quality of medical services, doctor recommendations). Subsequently, we will categorize these factors into levels to clarify their role and positions in patients' perception of recurrence risk. Finally, we will summarize and present the influencing factors and systematically present how each level impacted risk perception and the pathways of influence (Table 2).

Additional data will be extracted from interventions or qualitative studies. For the intervention studies, the intervention content and presence of a control group will be extracted. For qualitative studies, data collection methods, analysis methods, and research findings will be extracted. Data will be independently extracted by two authors, strictly following a pre-made datasheet. Any discrepancies during the data extraction process will be resolved through discussions among the review team. Original authors will be contacted to obtain any missing data.

**Collating, summarizing, and reporting the results.**    Owing to the exploratory nature of this scoping review, we will employ appropriate data synthesis methods for the existing data. Both qualitative and quantitative methods will be used for analysis, synthesis, and presentation. Tables, charts, diagrams, or flowcharts will be used as appropriate to present the extracted variables. Narrative and thematic analysis will be utilized to elucidate the substantive results. A

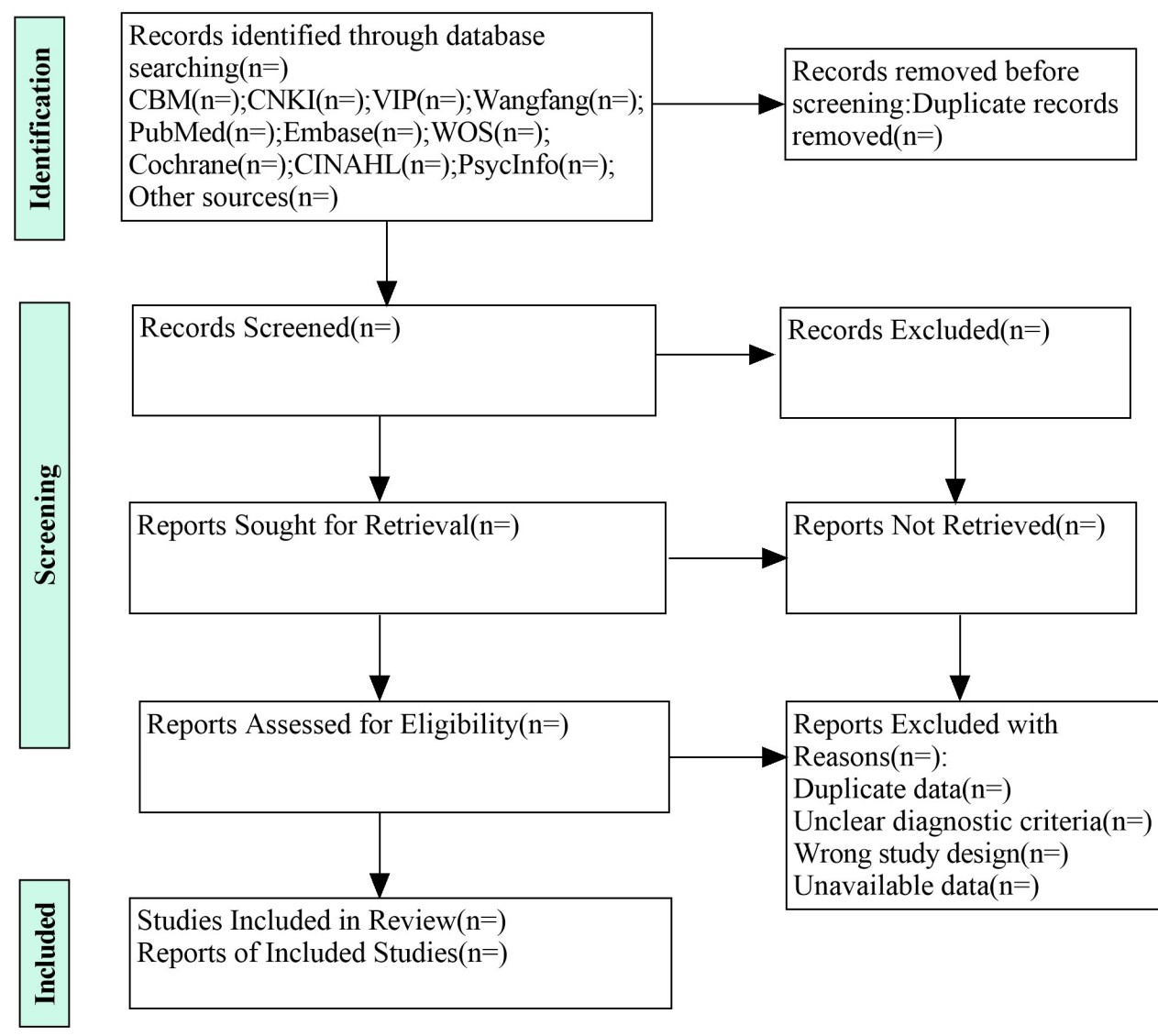

**Fig 1. PRISMA flow diagram.**

robust discussion will be based on a clear analysis of the results relevant to the review questions. Additionally, other issues of interest that may arise during the review process will be discussed, and summarizing these will lead to effective conclusions and related recommendations. Based on existing research, gaps in literature and potential directions for future research will be identified.

Critical appraisals are typically not recommended for scoping assessments. Given the exploratory and descriptive nature of our scoping review, we will not perform a critical

**Table 2. Data extraction instrument.**

| Author | Year of publication | Country/ region | Research target/sample size | Study method/ design | Research themes | General condition of the patient | Data analysis method | Assessment tools | Perceived level of recurrence risk | Influencing factors |
|--------|--------|--------|--------|--------|--------|--------|--------|--------|--------|--------|
| | | | | | | | | | | |

appraisal of individual evidence sources nor will we assess the quality of evidence in the included records.

Nevertheless, we include the limitations of this review.

## Ethics and dissemination

Finally, after data extraction, organization, and summary, we will report our scoping review findings according to the PRISMA-ScR guidelines [28]. These findings will be presented in a manner consistent with the objectives of the scoping review.

## Strengths and limitations

1. This will be the first review to examine existing research on stroke patients' perception of recurrence risk.

2. This review will employ a rigorous literature search strategy.

3. Review results will be reported according to the Preferred Reporting Items for Systematic Reviews and Meta-Analyses extension for Scoping Reviews (PRISMA-ScR) guidelines to ensure methodological rigor.

4. Only studies published in English and Chinese will be included, which may potentially introduce a language bias as studies in other languages might be overlooked.

5. This scoping review will not include a meta-analysis nor assess the quality of the included studies.

## Supporting information

**S1 Checklist. PRISMA-P (Preferred Reporting Items for Systematic review and Meta-Analysis Protocols) 2015 checklist: Recommended items to address in a systematic review protocol\*.**
(DOCX)

## Acknowledgments

We would like to thank Editage (www.editage.cn) for English language editing.

## Author Contributions

**Conceptualization:** Shanshan Zhu, Xin Guo, Dingding Li, Yage Shi.

**Data curation:** Shanshan Zhu, Meiqi Xu, Shuaiyou Wang.

**Formal analysis:** Meiqi Xu.

**Funding acquisition:** Huimin Zhang.

**Methodology:** Xueting Sun.

**Project administration:** Hongru Wang, Huimin Zhang.

**Writing – original draft:** Shanshan Zhu, Chenjun Liu.

**Writing – review & editing:** Dingding Li, Hongru Wang, Huimin Zhang.

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
