## [Decision Letter · Decision Letter 0]

6 Aug 2024

PONE-D-24-20180Research on Stroke Patients' Perception of Recurrence Risk：A Scoping Review ProtocolPLOS ONE

Dear Dr. Zhang,

Thank you for submitting your manuscript to PLOS ONE. After careful consideration, we feel that it has merit but does not fully meet PLOS ONE’s publication criteria as it currently stands. Therefore, we invite you to submit a revised version of the manuscript that addresses the points raised during the review process.

We look forward to receiving your revised manuscript.

Kind regards,

Apurva kumar Pandya, PhD

Academic Editor

PLOS ONE

Journal Requirements:

2. Thank you for stating the following financial disclosure: "Funding: This work is supported by the [Key Scientific Research Project of Higher Education Institutions of Henan Province] (grant number: 23B320002), [Henan Province Higher Education Teaching Reform Research and Practice Project] (grant number: 2023SJGLX236Y), and [Henan Province Graduate Education Reform and Quality Improvement Project] (grant number: YJS2024JC27)."

Additional Editor Comments:

The manuscript is well written; however, not acceptable in present form. Minor revision is suggested.

Reviewers' comments:

Reviewer's Responses to Questions

**Comments to the Author**

1. Does the manuscript provide a valid rationale for the proposed study, with clearly identified and justified research questions?

Reviewer #1: Yes

2. Is the protocol technically sound and planned in a manner that will lead to a meaningful outcome and allow testing the stated hypotheses?

Reviewer #1: Yes

3. Is the methodology feasible and described in sufficient detail to allow the work to be replicable?

Reviewer #1: Yes

4. Have the authors described where all data underlying the findings will be made available when the study is complete?

Reviewer #1: Yes

5. Is the manuscript presented in an intelligible fashion and written in standard English?

Reviewer #1: Yes

6. Review Comments to the Author

You may also provide optional suggestions and comments to authors that they might find helpful in planning their study.

Reviewer #1: The Protocol is explained in sufficient detail. However, in the section of data collection tool, there are certain typographical errors which needs to be corrected. At places, the tenses of the sentences are changing too frequently which can be rechecked. The methodology of how the protocol will help identify the influencing factors is not clear and hence can be described in more details. Overall, the manuscript can be accepted with minor revisions.

7. PLOS authors have the option to publish the peer review history of their article (what does this mean?). If published, this will include your full peer review and any attached files.

Reviewer #1: No

---

## [Author Response · Author response to Decision Letter 0]

14 Sep 2024

Response to Reviewers

Dear Dr. Apurva Kumar Pandya and Reviewers,

We would like to thank you for the thoughtful and constructive feedback on our manuscript entitled “Research on Stroke Patients' Perception of Recurrence Risk: A Scoping Review Protocol” (PONE-D-24-20180). We have carefully considered the comments from the academic editor and reviewer(s) and have revised the manuscript accordingly. Below, we provide a detailed response to each comment and explain the changes we have made in the revised version. All changes are highlighted in the manuscript using "Track Changes."

Reviewer #1 Comments:

Comment 1: The Protocol is explained in sufficient detail. However, in the section of data collection tool, there are certain typographical errors which need to be corrected.

Response: Thank you for pointing this out. We have carefully reviewed the entire manuscript and corrected all typographical errors, particularly in the section related to the data collection tool. These corrections are highlighted in the revised manuscript (see page 9, line 187 for the specific changes).

Comment 2: At places, the tenses of the sentences are changing too frequently which can be rechecked.

Response: We appreciate your suggestion regarding the consistency of tenses. We have thoroughly reviewed the manuscript and adjusted the tenses to maintain consistency throughout the text. The changes have been implemented and are visible in the revised manuscript.

Comment 3: The methodology of how the protocol will help identify the influencing factors is not clear and hence can be described in more detail.

Response: Thank you for this valuable feedback. We have added further clarification in the methods section to explain how the protocol will help identify influencing factors related to stroke patients' perception of recurrence risk. We have expanded on the description of the analysis framework and provided more specific details on the steps we will follow (see page 8, line 178-185 for the updated text).

Editor's Additional Requirements:

Comment 4 (Financial Disclosure): Please state what role the funders took in the study. If the funders had no role, please state: "The funders had no role in study design, data collection and analysis, decision to publish, or preparation of the manuscript."

Response: We confirm that the funders had no role in study design, data collection and analysis, decision to publish, or preparation of the manuscript. We have included the following statement in the cover letter and updated the financial disclosure accordingly: "The funders had no role in study design, data collection and analysis, decision to publish, or preparation of the manuscript."

Comment 5 (Data Availability): We strongly recommend all authors decide on a data sharing plan before acceptance. Please revise your data availability statement if necessary.

Response: We agree with the recommendation and have outlined a clear data sharing plan. Upon acceptance, all relevant data will be made available through a public repository, in accordance with PLOS ONE's open data policy. 

Comment 6 (Supporting Information): Please include captions for your Supporting Information files at the end of your manuscript, and update any in-text citations to match accordingly.

Response: We have reviewed and updated the Supporting Information section to include the appropriate captions. 

Comment 7 (References): Please review your reference list to ensure that it is complete and correct. If you have cited papers that have been retracted, please include the rationale for doing so or replace them with relevant current references.

Response: We have thoroughly reviewed our reference list and confirmed that all references are current and accurate. No retracted papers have been cited. If future revisions are necessary, we will update the reference list accordingly.

We hope that these revisions have addressed all concerns raised by the reviewer(s) and editor. Please let us know if further clarifications or changes are needed. We look forward to your feedback and appreciate your continued consideration of our manuscript.

Sincerely,

Huimin Zhang

[September 14th 2024]

Corresponding Author on behalf of all authors

---

## [Editor Report · Decision Letter 1]

2 Oct 2024

Research on Stroke Patients' Perception of Recurrence Risk：A Scoping Review Protocol

PONE-D-24-20180R1

Dear Dr. Zhang,

We’re pleased to inform you that your manuscript has been judged scientifically suitable for publication and will be formally accepted for publication once it meets all outstanding technical requirements.

Kind regards,

Apurva kumar Pandya, PhD

Academic Editor

PLOS ONE

Additional Editor Comments (optional):

Authors have addressed reviewers comments, and now the manuscript is acceptable.
---

## [Editor Report · Acceptance letter]

7 Oct 2024

PONE-D-24-20180R1 

PLOS ONE

Dear Dr. Zhang, 

I'm pleased to inform you that your manuscript has been deemed suitable for publication in PLOS ONE. Congratulations! Your manuscript is now being handed over to our production team.

Kind regards, 

on behalf of

Dr. Apurva kumar Pandya 

Academic Editor

PLOS ONE